



# Signatures of gravity wave-induced instabilities in balloon lidar soundings of polar mesospheric clouds

Natalie Kaifler[1], Bernd Kaifler[1], Markus Rapp[1], and David C. Fritts[2]

[1]Institute of Atmospheric Physics, German Aerospace Center, Oberpfaffenhofen, Germany
[2]GATS, Boulder, CO, USA

**Correspondence:** Natalie Kaifler (natalie.kaifler@dlr.de)

**Abstract.** The Balloon Lidar Experiment (BOLIDE) which was part of the PMC Turbo balloon mission has captured near-vertical profiles of polar mesospheric clouds (PMC) during a 6-day flight along the Arctic circle in July 2018. The high-resolution soundings (20 m vertical and 10 s temporal resolution) reveal highly structured layers with large gradients in volume backscatter coefficient. We systematically screen the BOLIDE dataset for small-scale variability by assessing these gradients at
high resolution. We find longer tails of the probability density distributions of these gradients compared to a normal distribution, indicating intermittent behaviour. The high occurrence rate of large gradients is assessed in relation to the 15-min-averaged layer brightness and the spectral power of short-period (5–62 min) gravity waves based on PMC layer altitude variations. We find that variability on small scales occurs during weak, moderate and strong gravity wave activity. Layers with below-average brightness are less likely to show small-scale variability in conditions of strong gravity wave activity. We present and discuss
signatures of this small-scale variability and possibly related dynamical processes, and identify potential cases for future case studies and modelling efforts.

## 1   Introduction

The intricate small-scale structure of noctilucent or polar mesospheric clouds (PMC) has been attributed to the action of gravity waves very early on (Witt, 1962). To this day, the strong backscatter of the ice particles that PMC are made of represents an
unique opportunity to observe dynamical processes in the upper mesosphere. High-resolution observations of PMC are obtained using both active and passive remote sensing instruments that are based on ground or carried by satellite, aircraft or balloon (e.g. Baumgarten and Fritts, 2014; Gao et al., 2018; Miller et al., 2015; Reimuller et al., 2011; Schäfer et al., 2020). Lidar instruments allow for high vertical resolution down to few meters, revealing various stages in the approach to, or attainment of, gravity wave breaking and other nonlinear dynamics made visible by the PMC layer. However, a comprehensive physical
interpretation of the waves and structures inside the PMC layer observed by lidar is limited by the sparseness of the horizontal sampling. Complementary spatial information can be provided by e.g. cameras, but opportunities in terms of time and location for observations from ground are scarce, as lidars operate best in darkness, but cameras can observe PMC in twilight only. Hence, both camera and lidar instruments have to be operated at significant distance for common-volume observations and simultaneous observations depend on suitable conditions at two places (Baumgarten et al., 2009). This limitation is overcome





by lifting a single payload including a lidar instrument and cameras to the upper stratosphere, where the sky is always dark and the view unhindered by tropospheric clouds. This was accomplished first by the PMC Turbo mission that observed the PMC layer along the Arctic circle during a 6-day flight in July 2018 (Fritts et al., 2019).

The effect of gravity waves and processes associated with their breaking on the PMC layer is not fully understood to date, in part due to the wide range of scales, the often unknown state of the background atmosphere and the microphysical evolution of
PMC particles in this multi-scale environment. Whether stratospheric gravity wave activity influences the occurrence of PMC or not, has been answered differently at different locations (Thayer et al., 2003; Innis et al., 2008; Chu et al., 2009). Gravity wave activity based on upper mesospheric radar measurements was not found to be correlated with the occurrence of PMC at a specific Arctic site (Wilms et al., 2013). However, Chandran et al. (2012) deduced from the global CIPS dataset that regions of strong gravity activity, both long- and short-period, suffer from reduced PMC brightness on average. This finding is supported
by a number of modelling studies using microphysical models that showed that gravity waves generally act to destroy PMC (Jensen and Thomas, 1994; Rapp et al., 2002; Dong et al., 2021). By means of the induced variations in both temperature and altitude, ice particles are ultimately exposed to unfavourable conditions, resulting in their rapid sublimation. The asymmetry in growth and sublimation rates ensures that particles are more likely to be destroyed than to grow large enough in size to be observable, and that particles once sufficiently reduced in diameter are unlikely to re-grow to observable sizes for a significant
amount of time. The outcome may depend on the period of gravity waves modulating the environment in the vicinity of PMC. Rapp et al. (2002) found a reduction of PMC brightness for short-period gravity waves below 6.5 h only. In three-dimensional simulations, Dong et al. (2021) observed the sublimation of a modelled PMC layer caused by breaking gravity waves within 30 min, leading to the formation of holes (or voids) which resemble structures observed in PMC satellite images (Thurairajah et al., 2013). At timescales of minutes, the PMC layer can be considered a passive tracer of the dynamic processes (Fritts et al.,
1993; Dong et al., 2021). As visual PMC frequently display small-scale structures attributed to gravity waves and processes associated with their breaking, high-resolution observations of the PMC layer are ideally suited to study the statistics and pathways of energy transfer from gravity waves down to turbulent scales (Warhaft, 2000; Shraiman and Siggia, 2000).

Among the millions of images acquired during the PMC Turbo mission were intriguing displays of small-scale vortex rings (Geach et al., 2020), Kelvin-Helmholtz instabilities (Kjellstrand et al.) and mesospheric bores (Fritts et al., 2020). The selection
of these events was mainly based on the interpretation of the images that were postprocessed to enhance small-scale structure (Kjellstrand et al., 2020) and make up only the tip of an iceberg of potentially interesting observations. Colocated lidar data have in all cases revealed significant small-scale structures with signatures that are likely inherent to the dynamic processes studied. To foster further analysis and interpretation of this dataset and others, we here aim to provide statistics and a systematic review of such patterns. We will employ a general, easy-to-implement method for identifying features related to quick changes
in PMC brightness on scales of $40\,\mathrm{m}$ and $20\,\mathrm{s}$ that represent processes related to dynamical instabilities following the breaking of gravity waves. The results can be used to select promising subsets of the PMC Turbo dataset for subsequent case studies, aid the interpretation of similar signatures in other lidar datasets, and serve as a reference for future modelling of the effect of gravity waves on PMC layers. As stated by Fritts et al. (2019), such observations have the potential to confirm or constrain modelling results, or discover processes that are yet unknown. In addition, we present a statistical analysis of the occurrences





of such patterns with regard to ambient conditions. Gravity wave activity and layer brightness can be derived from the lidar data and thus provide the chance to relate the occurrence of small-scale structures, that are related to GW breaking or other nonlinear dynamics, to the strength of short-period gravity wave actitivity and the mean layer brightness. The intention is to find out whether the studied events arise during periods of intense gravity wave activity or rather during quiet times, and whether this goes along with enhanced or reduced PMC brightness. This study thus also aims to complement the modelling study of

Dong et al. (2021) with a comprehensive observational dataset. The step beyond previous analysis of observational data roots in the high temporal and spatial resolution of the PMC Turbo lidar dataset that facilitates the detection of turbulent-like features and instabilities that arise as an effect of breaking gravity waves. Intentionally, we focus this study on the PMC lidar dataset of PMC Turbo and will make use of the associated images in subsequent case studies identified by this work.

## 2 Method

### 2.1 Instrument and data

The Balloon Lidar Experiment BOLIDE is a Rayleigh backscatter lidar designed to operate onboard a stratospheric long-duration balloon at an altitude of ∼40 km at polar latitudes (Kaifler et al., 2020). During the 6-day flight of PMC Turbo along the Arctic circle from Sweden to Canada in July 2018 almost 50 h of high-resolution PMC soundings were recorded (Kaifler et al., 2022; Fritts et al., 2019). PMC are detected relative to a standard atmosphere density profile as described in detail by

Kaifler et al. (2022). In short, the atmospheric return signal from the 28° off-zenith tilted 4.2 W laser beam was collected by a 0.5 m diameter receiving telescope and detected by an avalanche photodiode operated in a single-photon counting mode. The resulting photon signals were timestamped with nanosecond resolution allowing for flexible choices in binning the lidar data in time and range during postprocession. In data analysis, the measurements obtained along the line of sight of the laser beam are converted to altitude. The relevant physical quantity measured by the lidar is the volume backscatter coefficient $\beta$, a

measure for PMC brightness influenced by both the size and number density of ice particles within the observed volume. This work makes use of lidar data at 20 m vertical and 10 s temporal resolution. Although higher resolutions are possible during bright displays with a high signal-to-noise ratio, these parameters are a suitable choice for the type of analysis carried out in this study. We include values of $\beta$ that are 2.5 standard deviations above the background and discard profiles where the vertical sum of $\ln \beta$ in units of $10^{-10}/\text{m/sr}$ is below a threshold value of 30. The intention is to reject sporadic detections with low

significance that cannot be attributed to a coherent PMC layer. The threshold is chosen such that the estimate of the mean layer altitude is sufficiently smooth for the subsequent spectral analysis. 41.9 h of PMC detections remain for the analysis.

### 2.2 Gradients in high-resolution data

Volume backscatter coefficients of PMC, $\beta$, follow an exponential distribution (Berger et al., 2019; Kaifler et al., 2022). It is therefore convenient to plot and analyze $\beta$ on a logarithmic scale. On a linear scale, large $\beta$ dominate, and the details of the

variations at smaller $\beta$ are lost. As a measure for small-scale variability, we evaluate the gradients $\partial/\partial z \ln \beta$ and $\partial/\partial t \ln \beta$,





abbreviated $\partial_z\beta$ and $\partial_t\beta$ in altitude $z$ and time $t$. Gradients are widely used in data analysis, e.g. in the analysis of topographic lidar data of complex terrain for the derivation of high-resolution digital elevation models (Szypuła, 2019). We compute the derivatives using a three-point (quadratic) Lagrangian interpolation at a resolution of 20 m and 10 s. Large values of $\partial_z\beta$ indicate distinct layer edges and highlight thin and bright multiple layers. Moving from low to high altitudes, positive (negative)

values of $\partial_z\beta$ mark lower (upper) layer boundaries. Large values of $\partial_t\beta$ contrast vertical motions of the PMC layers presumably induced by vertical winds, or spatial patterns that are horizontally advected through the lidar's field of view. We will later select large absolute values of $\partial_z\beta$ and $\partial_t\beta$ to identify periods with intense small-scale variability.

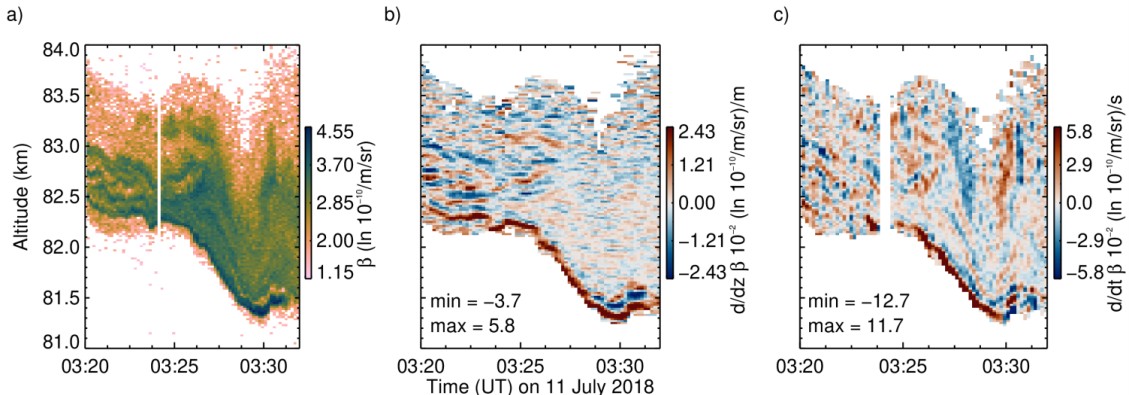

**Figure 1.** (a) Volume backscatter coefficients $\beta$ between 3:20 and 3:35 UT on 11 July 2018, and (b) gradients in altitude, highlighting vertically-stacked layers before 3:27 UT, and (c) gradients in time, clearly showing the quick descent and following ascent of the PMC layer around 3:29 UT. The resolution of the data analzyed and shown in (a) is 20 m and 10 s, the gradients shown in (b) and (c) have been smoothed to 40 m and 20 s, respectively.

We illustrate the application of the gradient analysis in both dimensions with an event observed on 11 July between 3:20 UT and 3:33 UT. We will show later that this particular event exhibits the largest gradients in the PMC Turbo BOLIDE dataset.

Fig. 1 shows volume backscatter coefficients as measured by the lidar as a function of altitude and time, as well as both the vertical and temporal gradients. To enhance the visibility of the plots, we smooth the gradients between adjacent profiles, i.e. to 20 s for $\partial_z\beta$ and to 40 m for $\partial_t\beta$. At 3:20 UT a wide layer above 82 km altitude with pronounced internal structure including at least four narrow, closely-spaced sublayers below 100 m width are observed. At 3:26 UT, a sudden, steep descent of the PMC layer occurs. Notable is the bright and distinct lower boundary that descends by 1 km from 82.2 km to 81.2 km. Best visible

in Fig. 1c, the upper boundary is subjected to a similar downward motion, yet starting 3 min later and descending quicker and deeper (area with blue color). A corresponding increase in layer altitude follows shortly after (red color at 3:30 UT). At the minimum altitude at 3:30 UT, the layer brightness quickly increases to $\beta = 4.37 \times \ln 10^{-10}/\text{m/sr}$ for a duration of 1 min. This goes along with large gradients of $\partial_z\beta = 5.8 \times 10^{-2} \ln(10^{-10}/\text{m/sr})/\text{m}$ and $\partial_t\beta = -12.7 \times 10^{-2} \ln(10^{-10}/\text{m/sr})/\text{s}$.





### 2.3 Background environment

To define a proxy for gravity wave activity we follow Geach et al. (2020) who studied vortex rings of $\approx 5$ km diameter generated by the breaking of a high-frequency gravity wave. Its signature in lidar data proved to be sinusoidal oscillations of the PMC layer with a period of 670 s. Based on this result, we construct a proxy $P_{\mathrm{GW}}$ for wave activity from PMC layer altitude variations. We start with the time series of the $\ln\beta$-weighted mean PMC layer altitude $z_c$ that is filled with linear interpolations at times of below-threshold or no PMC detections. $z_c$ is then spectrally decomposed using Morlet wavelets,
which are now widely used in analysis of atmospheric data and are considered an optimal choice regarding the periodicity and localization of wave structures in PMC (e.g. Rong et al., 2018). To retain high-frequency gravity waves above the buoyancy frequency only, we select the spectral range of 5–62 min period. $P_{\mathrm{GW}}$ is then finally determined as the averaged spectral power within this band. It is inherently smoothed by the wavelet analysis and can thus be used as a representation of the gravity wave background.

As a proxy $P_\beta$ for the mean PMC layer brightness we employ vertically integrated volume backscatter coefficients smoothed by a 15 min running mean. The vertical integration facilitates comparisons with camera and visual observations, which observe accumulated brightness along the line of sight, and is insensitive to effects of convergence; for example, a wide, dim layer has the same integrated brightness as a thin, bright layer. The smoothing to lower resolution effectively removes brightness variations at scales below 15 min, and thus separates this quantity from the small-scale variability assessed by the gradient
analysis. As a consequence of the vertical integration and temporal smoothing, as with $P_{\mathrm{GW}}$, the number of independent data points in $P_\beta$ is reduced compared to the gradients evaluated on high-resolution data. For illustration of $P_{\mathrm{GW}}$ and $P_\beta$, we show a larger part of the PMC observed on 11 July 2018 including the part around 3:30 UT we selected to demonstrate the gradient analysis in the previous section (Fig. 2a). The mean PMC altitude $z_c$ (Fig. 2b, black curve) is a good proxy for layer altitude and the spectrally filtered time series effectively captures the high-frequency part of the motion (Fig. 2b, blue curve). $P_{\mathrm{GW}}$ is
maximum at 4 UT due to the change in altitude of almost 2 km in the selected spectral band (blue in Fig. 2c). Maxima in $P_\beta$ are found around 5:10 UT and 6 UT when the layer is locally bright or wide (brown in Fig. 2c).

### 3 Results

At a resolution of 20 m and 10 s, the BOLIDE dataset includes $1.06 \times 10^6$ independant measurements with sufficient signal-to-noise ratio of PMC volume backscatter coefficient. A number of $n_v = 672921$ have neighbours that allow for the calculation
of both gradients $\partial_z\beta$ and $\partial_t\beta$. They occur within $n_p = 13255$ independent vertical profiles totalling 36.8 h. Fig. 3ab show normalized probability density functions of all gradients $\partial_t\beta$ and $\partial_z\beta$ derived from measured data. The distributions have approximately zero mean and extend to $\partial_z\beta = 6.0 \times 10^{-2}\ln(10^{-10}/\mathrm{m/sr})/\mathrm{m}$ and $\partial_t\beta = 13.4 \times 10^{-2}\ln(10^{-10}/\mathrm{m/sr})/\mathrm{s}$. Gradients of this magnitude mean that the locally brightest PMC volume of $4.67 \times \ln 10^{-10}/\mathrm{m/sr}$ can potentially emerge or fade within $4.67/0.060 = 78\,\mathrm{m}$ in the vertical or $4.67/0.134 = 35\,\mathrm{s}$ in time. The maximum gradient value is thus close to the limits given
by the resolution and smoothing of the gradients.

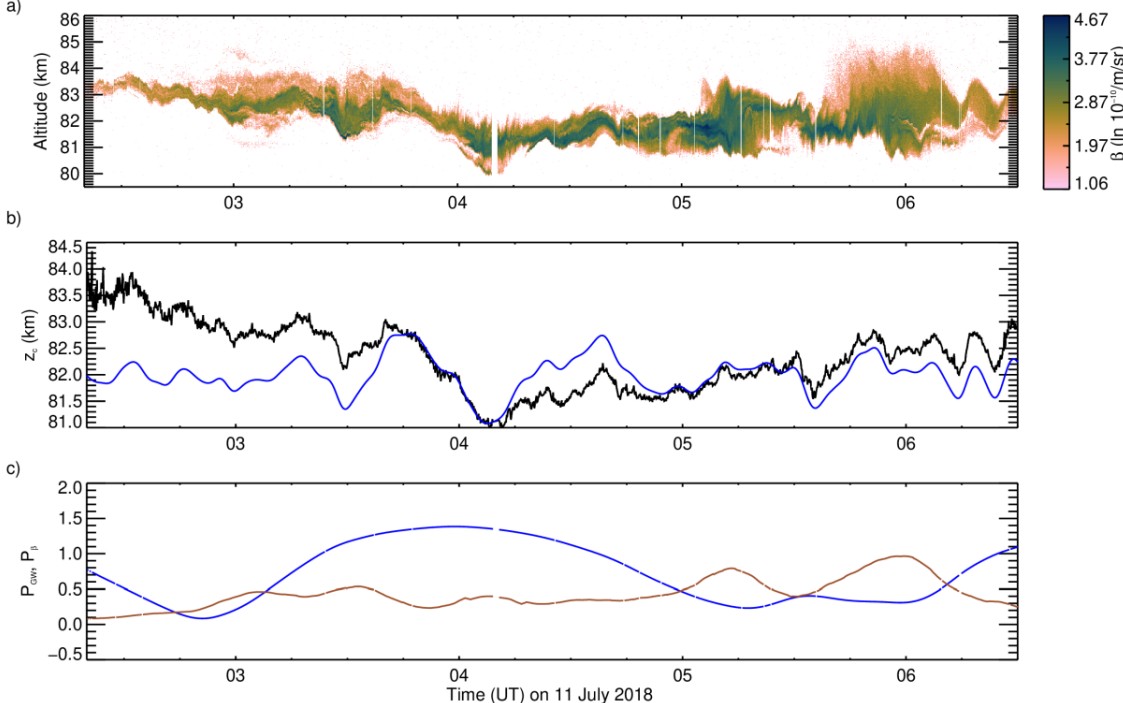

**Figure 2.** (a) Volume backscatter coefficients of the PMC layer on 11 July 2018, 2:20–6:30 UT. (b) PMC layer altitude $z_c$ (black) and altitude variations reconstructed from a range of filtered scales corresponding to periods between 5-62 min (blue), (c) scale-averaged spectral power $P_{\mathrm{GW}}$ (blue, units of ln km$^2$) within 5-62 min period as a proxy for short-period gravity wave activity and 15-min-smoothed vertically integrated brightness $P_\beta$ (brown, units of ln $10^{-6}$/sr) as a proxy for background PMC layer brightness.

The moments of the distributions of $\partial_z\beta$ and $\partial_t\beta$ are listed in Tab. 1. The standard deviations are $\sigma_z = 0.79\times10^{-2}\ln(10^{-10}/\mathrm{m/sr})/\mathrm{m}$ and $\sigma_t = 1.66\times10^{-2}\ln(10^{-10}/\mathrm{m/sr})/\mathrm{s}$, respectively. The distributions are skewed to positive values, especially $\partial_z\beta$. This indicates steeper lower boundaries of PMC layers or respective sublayers compared to top boundaries. The kurtosis is positive, meaning that the distributions have longer tails than the normal distribution. For comparison, we plot Gaussian distributions with the same mean and standard deviations in Fig. 3ab as dotted lines. This is an indication for intermittent behaviour.

To further characterize the distributions of $\partial_z\beta$ and $\partial_t\beta$ with regard to the occurrence of large gradients, we shift a threshold $l\sigma$ with $\sigma$ being the standard deviation of the data set and $l = 0...5$. $l = 2$ is marked by vertical lines in Fig. 3ab as an example. For $l = 2$, 5.3 % of the $\partial_z\beta$ ($\partial_t\beta$) values are found in the tail of the respective distribution (Fig. 3ab), while 9.7 % of all PMC detections exhibit gradients that are within the tail of either distribution. The latter quantity is shown in Fig. 3c as a function of $l$ as solid curve. The dashed curve shows the percentage of vertical profile that exhibit gradients within the tail of either distribution. The result that 9.7 % of all gradients that are found in the tails for $l = 2$ are actually distributed across 96.6 % of vertical profiles means that large gradients occur frequently and that smooth and unperturbed PMC layers are rare.





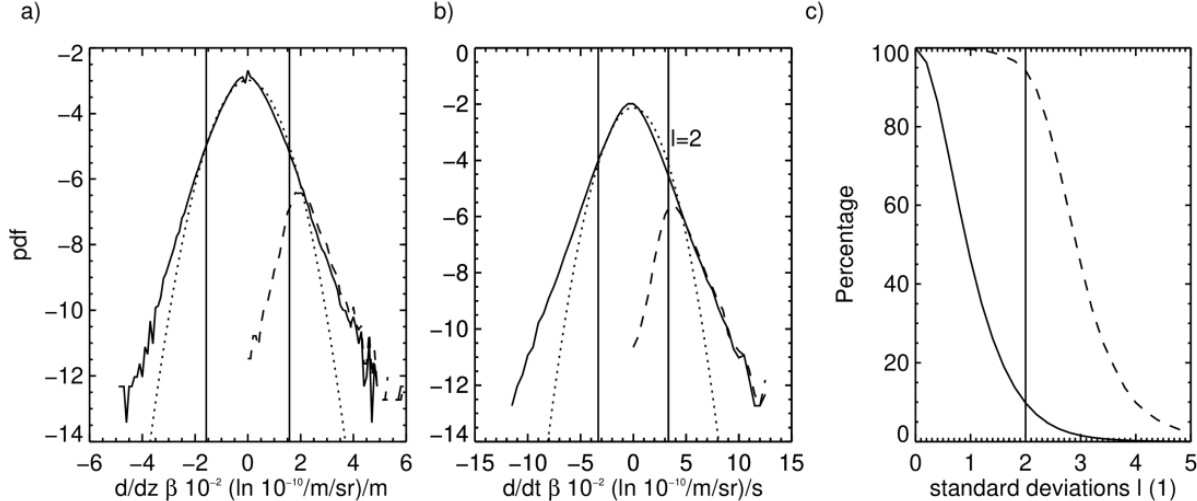

**Figure 3.** Probability density function of a) vertical brightness gradients $\partial_z \beta$ and b) temporal brightness gradients $\partial_t \beta$. The solid line shows the distribution of all values normalized to unity. The dotted lines shows the Gaussian distribution of the same mean and standard deviation. The dashed line shows the maximum of the absolute gradient value per profile. The vertical lines are drawn at $\pm 2\sigma$, and include 94.7 % of all values. c) Percentage of PMC values (black line) and profiles (dashed line) with $|\partial_z \beta| > l\sigma_z$ or $|\partial_t \beta| > l\sigma_t$ depending on the threshold $l$, where $l = 2$ is again marked. 9.7 % of all gradient values are above $2\sigma$, and 96.6 % of all PMC profiles contain gradients of either $|\partial_z \beta| > 2\sigma_z$ or $|\partial_t \beta| > 2\sigma_t$.

**Table 1.** Statistical moments of $\partial_z \beta$ in units of $10^{-2} \ln(10^{-10}/\text{m/sr})/\text{m}$ and $\partial_t \beta$ in units of $10^{-2} \ln(10^{-10}/\text{m/sr})/\text{s}$. A raw kurtosis of 3 that is expected for a Gaussian distribution was subtracted.

|  | mean | standard deviation | skewness | kurtosis |
|---|---|---|---|---|
| $\partial_z \beta$ | 0.00 | 0.79 | 0.09 | 1.04 |
| $\partial_t \beta$ | 0.00 | 1.66 | 0.04 | 1.47 |

The mean $P_{\text{GW}}$ of all PMC observations within the BOLIDE dataset is $0.76 \pm 0.77 \ln \text{km}^2$, corresponding to a PMC layer altitude variability of $\sqrt{\exp(0.76)} = 1.46$ km in the 5–62 min spectral band. The minimum value is -2.1 and corresponds to a variability of 350 m while the maximum value is 2.26 with a variability of 3.1 km. The mean $P_\beta$ is $0.29 \pm 0.18 \ln 10^{-6}/\text{sr}$ and values range between 0.02 and $0.96 \times \ln 10^{-6}/\text{sr}$. We divide the PMC profiles into four categories relative to these mean values $\overline{P_{\text{GW}}}$ and $\overline{P_\beta}$ in order to assess the occurrence rate of large gradients in brighter or dimmer PMC layers exposed to weak or strong gravity wave activity. The selection criteria are defined in Tab. 2. Though desireable, a finer division is not implemented because of the limited number of independent values due to the smoothing of $P_{\text{GW}}$ and $P_\beta$. The fractions of gradient values per category as a function of the threshold $l$ are shown in Fig. 4, where again to the left for $l = 0$ all profiles are included, and to the right only those with larger gradients in either $\partial_t \beta$ or $\partial_z \beta$ remain. We find that dimmer layers with strong gravity wave





activity (category B in Fig. 4) are most common, followed by brighter layers with strong gravity wave activity (category D). As the statistics is narrowed to the largest-gradient profiles, category-D-layers make up for the largest share. The PMC layers belonging to the most common category B are however unlikely to exhibit large gradients compared to layers of the other

categories.

**Table 2.** Criteria for classification of PMC layers (dim or bright and strong or weak gravity wave activity) according to $\overline{P_\beta}$ and $\overline{P_\beta}$).

|  | A | B | C | D |
|---|---|---|---|---|
| $P_\beta(\ln 10^{-6}/\mathrm{sr})$ | $< 0.29$ | $< 0.29$ | $> 0.29$ | $> 0.29$ |
| $P_{\mathrm{GW}}(\ln \mathrm{km}^2)$ | $< 0.76$ | $> 0.76$ | $< 0.76$ | $> 0.76$ |

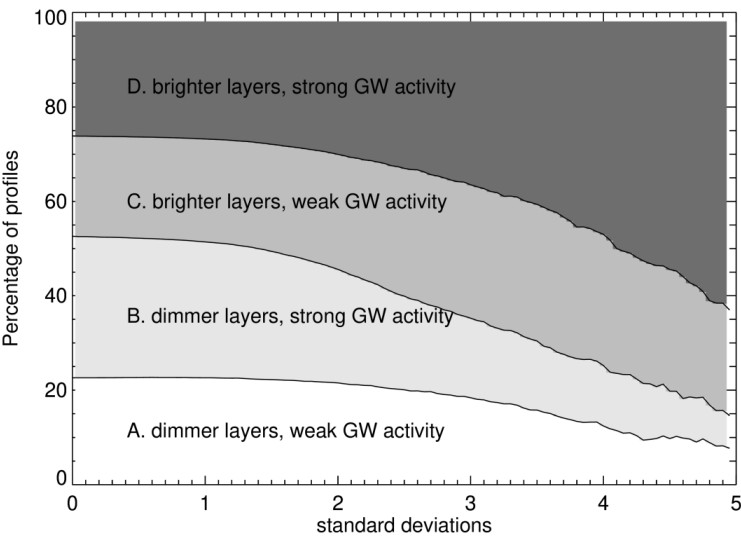

**Figure 4.** Percentage of PMC profiles with gradients $|\partial_z \beta| > l\sigma_z$ or $|\partial_t \beta| > l\sigma_z$ for the four categories defined in Tab. 2. For $l = 0$ all PMC profiles are included while at larger values of $l$ only profiles that exhibit increasingly larger gradients are considered.

## 4   Discussion

The PMC layer sections that contain mostly PMC profiles with gradients larger than $2\sigma$ exhibit all type of small-scale variability including vertical, oscillatory motions of bright, thin layers, internal structure of wider layers, multiple layers, and vertical displacements of PMC layers. Both large and small displacements are included. Although more sophisticated methods could

be developed to detect those signatures, we deem the gradient analysis as a suitable choice. It is an easy-to-implement and general method that is not tailored to specific patterns and is thus suitable for the detection of signatures of potentially previously unknown dynamics. We refrained from the application of wavelets, one- or two-dimensional, to assess the small-scale





variability of PMC as we did not want to make the assumption of an underlying periodicity but to allow for singular or irregular structures within the PMC layer at the smallest scales.

**Table 3.** PMC layer category and percentage of PMC detections or profiles with large gradients for the periods of nine major PMC events defined by Fritts et al. (2019, their Tab. 2). For reference, the data-set mean percentage of PMC detections with gradients above the $2\sigma$ threshold is 9.8 %, and for PMC profiles with gradients above the $4\sigma$ threshold is 9.9 % (from Fig. 3c).

| Date | Mean category based on $\overline{P_{\mathrm{GW}}}$ and $\overline{P_\beta}$ | $n_v$ for $|\partial_z\beta| > 2\sigma_z$ or $|\partial_t\beta| > 2\sigma_t$ | $n_p$ for $\max|\partial_z\beta| > 4\sigma_z$ or $\max|\partial_t\beta| > 4\sigma_t$ | Referenced subsets of data |
|---|---|---|---|---|
| 8 July 13–18 UT | A | 9.7 % | 0.8 % | |
| 9 July 1–4 UT | B | 12.2 % | 8.2 % | Fig. 8a |
| 9 July 12–19 UT | B | 14.1 % | 2.0 % | |
| 10 July 1:30–7 | B | 14.1 % | 11.6 % | Fig. 6a, Geach et al. (2020) |
| 10 July 17:30–22:30 | B | 14.8 % | 18.0 % | Fig. 5a, Fig. 6bc |
| 11 July 0:30–15:00 UT | C | 9.8 % | 13.6 % | Fig. 1, Fig. 2, Fig. 5b, Fig. 7a, Fig. 8bc |
| 11 July 21–24 UT | A | 10.5 % | 0.8 % | |
| 12 July 11–18 UT | C | 6.8 % | 9.6 % | Kjellstrand et al. |
| 13 July 6–16 UT | A | 9.8 % | 6.8 % | Fig. 5c, Fig. 7bcd, Fritts et al. (2020) |

The evaluation of the statistics of the gradients, in particular Fig. 3c, has established that such events are widespread whenever PMC is present. This is in agreement with the visual impression from both ground-based observations and the PMC Turbo images of highly variable PMC layers at small spatial scales. Fritts et al. (2019) divided the PMC Turbo dataset into eight major PMC events and gave an initial assessment of the relevant dynamics and likely impact, stating the magnitude of the forcing and the occurrences of gravity wave breaking, fronts or bores and Kelvin-Helmholtz instabilities. In Table 3 we complement this

list with results of our analysis, including category based on $P_{\mathrm{GW}}$ and $P_\beta$ and percentages of $n_v$ and $n_p$ in the tails as shown in Fig. 3c. The assessment by Fritts et al. (2019) of weak, moderate or strong gravity wave activity or forcing is reflected in our result for category. The events that stood out by initial visual inspections of PMC Turbo images and lidar data, some of which have already been analyzed for their dynamics, all show increased, and in most cases even very large gradients. The small-scale vortex rings studied by Geach et al. (2020) were observed on 10 July 2018 between 2:30–2:57 UT. Their signature

includes a rapid (1-min period) downward extension of comparably weak PMC backscatter by almost 1 km from the lower PMC boundary. The upper boundary, in contrast, was clearly defined and constant in altitude. This resulted in a large vertical gradient $\partial_z\beta$ of $4.4 \times 10^{-2}\ln(10^{-10}/\mathrm{m/sr})/\mathrm{m}$ at the top boundary, and the variation of the lower boundary is clearly reflected in quick successions of positive and negative values of $\partial_t\beta$ amounting to $\pm 9.6 \times 10^{-2}\ln(10^{-10}/\mathrm{m/sr})/\mathrm{s}$. This case was the only occurrence of small-scale vortex rings located at the lower PMC boundary in this clarity during PMC Turbo. Rapid os-

cillations in $\partial_t\beta$ at the top boundary or within the layer were also observed on 10 July 2018 at 3:17 UT and 11 July 2018 at 9:20 UT (not shown here). Also characterized by rapid changes in the temporal evolution are the Kelvin-Helmholtz instabilities





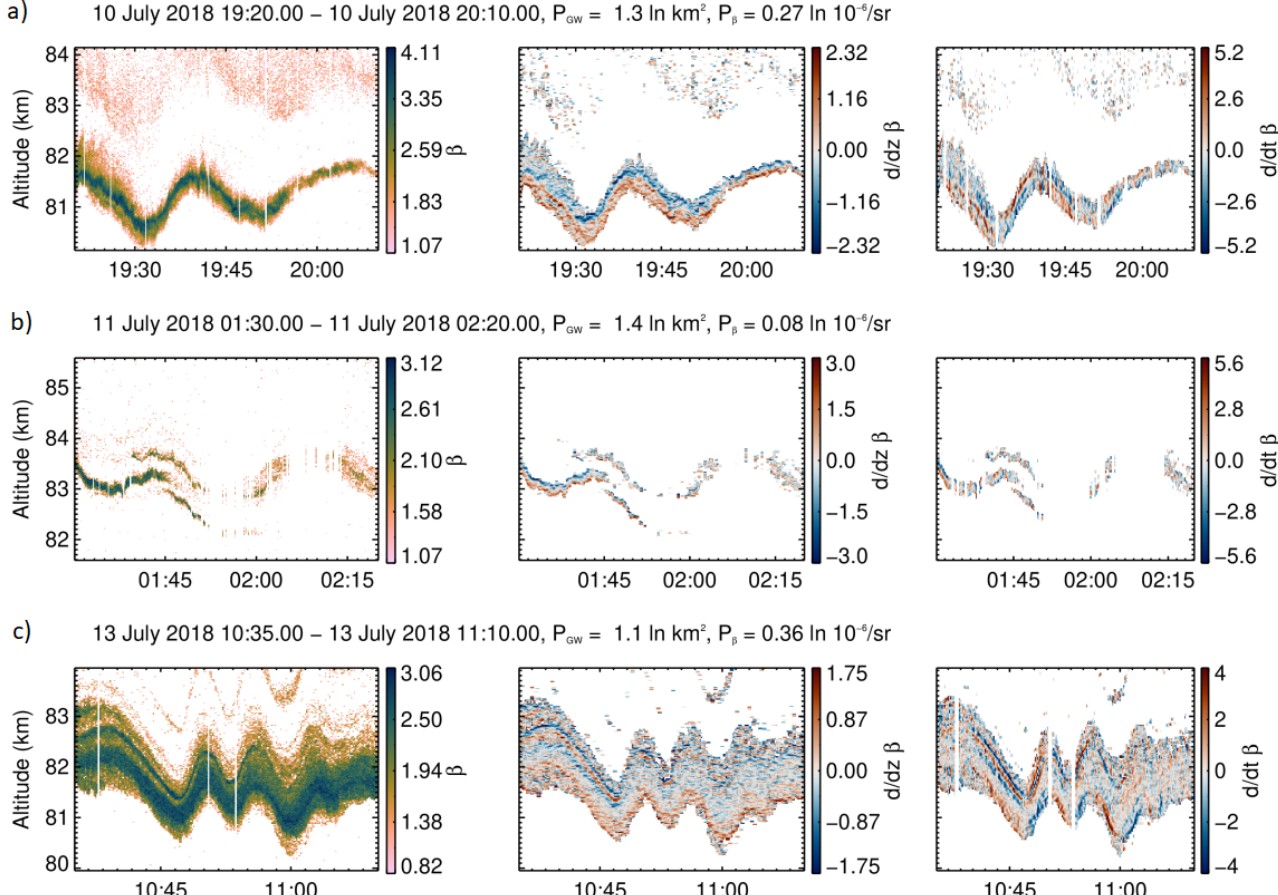

**Figure 5.** Examples for PMC layers with largely sinusoidal oscillations in altitude. The title line gives the start and end dates, as well as the mean $P_\beta$ and $P_{\mathrm{GW}}$ for this duration. The color scale includes the minimum and maximum of $\beta$ in units of $\ln 10^{-10}/\mathrm{m/sr}$ and two standard deviations of the gradient values (units of $10^{-2}(\ln 10^{-10}/\mathrm{m/sr})/\mathrm{m}$ and $10^{-2}(\ln 10^{-10}/\mathrm{m/sr})/\mathrm{s}$). All vertical axis extend 4.5 km in altitude and are centered around the mean $z_c$.

that occurred on 12 July at 13:30 UT that were studied in detail by Kjellstrand et al.. During this event, vertical displacements affect the whole PMC layer including multiple narrow sub-layers and a sharp lower boundary. The induced absolute gradients amount to $4.5 \times 10^{-2} \ln(10^{-10}/\mathrm{m/sr})/\mathrm{m}$ and $11.3 \times 10^{-2} \ln(10^{-10}/\mathrm{m/sr})/\mathrm{s}$ and are thus among the largest observed.

Similar signatures are visible in the aftermath of this specific event until 12 July 14:20 UT, as well as on 10 July between 18:45–19:20 UT, on 11 July between 3:55–6:00 UT (compare Fig. 8c) and on 11 July at 13:30 UT (not shown). Looking at the remaining dataset of PMC profiles with large gradients, the patterns can be associated with one or more of four groups:

1. PMC layers with largely sinusoidal undulations in altitude. The selected examples shown in Fig. 5 show undulations with periods between 5 min and 30 min and are thus likely induced by linear, short-period gravity waves. For the observation shown

in Fig. 5a, corresponding PMC Turbo images have confirmed a linear gravity wave of 20 min period and $\approx 50$ m/s phase speed.





Also $P_{\mathrm{GW}}$ is well above its mean, putting these layers in category B or D. This type of PMC layers exhibits large gradients for several reasons: First, the high definition of the layer edges leads to large vertical gradients where the layers emerge from or fade into the background. This is seen e.g. in $\partial_z\beta$ on 10 July 2018 at 19:20 – 20:10 UT (Fig. 5a). Second, due to the oscillatory motion, for the same reason also large temporal gradients are induced, e.g. on 13 July 2018 10:25–11:05 UT (Fig. 5c). But a
closer look reveals that those PMC layers are not smooth at smaller scales. Superposed to the larger-scale linear oscillation are small-scale perturbations at time scales of 1-2 min, which result in the detection of large gradients, e.g. around 19:40 UT on 10 July 2018 (Fig. 5a) and at 1:35 UT on 11 July 2018 (Fig. 5b). The PMC Turbo images for Fig. 5a, for example, show evidence for large- and small-scale vortex rings, embedded local instabilities and multi-scale breaking (not shown here). The example 13 July 2018 (Fig. 5c) highlights another common feature with very narrow-spaced (100 m or less), well-defined
sub-layers visible 10:35–10:48 UT. The brightest sub-layer starting at 82.6 km can be traced over several gravity-wave periods until around 11:00 UT, when it appears to dissolve. As in all examples of Fig. 5, multiple layers exist with interlayer spacing between 100 m and 2 km.

2. PMC layers indicating gravity wave breaking in response to gravity wave steepening. Examples of these very variable layer-types are shown in Fig. 6. In these cases, the amplitude of an undulation increases and the period shortens. This is followed
by a reduction of PMC brightness and/or the emergence of small-scale structures. For example around 5:33 UT on 10 July 2018 (Fig. 6a) a PMC layer perturbed by a large-amplitude gravity wave that reduced to 10 min period experiences a quick descent with a peak-to-peak amplitude of almost 3 km and $\beta$ values distributed over a wide range down to the detection limit, indicating complex instability dynamics. Interestingly, the situation seems analogous but mirrored in time around 5:48 UT except that here a double layer is affected. For a full interpretation of such cases, knowledge of the gravity wave propagation directions
and wind speeds relative to the lidar beam path across the PMC layer and the 2d-dimensional imaging of PMC structures are very helpful. Nevertheless this lidar observation seems to confirm theoretical results predicting that very strong gravity wave activity inducing large displacements significantly reduce PMC brightness close to or below the detection threshold (Dong et al., 2021). The time frame on 10 July between 17:45 and 19 UT is another example of a strong PMC layer modulated by a 10–30 min-period gravity wave that causes a multitude of strong instabilities including cusps, linked rings and "herringbone"
structures (see Fig. 11b of Fritts et al., 2019, at 18:15 UT and their discussion). The examples featured in Fig. 6 have the largest $P_{\mathrm{GW}}$ in the PMC Turbo dataset.

3. PMC layers that show transitions from weak or no gravity wave activity to major bore-like responses during few or tens of minutes. These type of layers suggest a rapid evolution or rapid advection of the leading edge of a nonlinear gravity wave packet. Examples are shown in Fig. 7. At 5:13 UT on 11 July 2018, the core of a PMC layer is both deflected upward and
downward by 2 km and 1 km, respectively. The corresponding imagery reveals a narrow and bright band which extends several hundred km across the camera field of view and moves through the lidar beam at 5:13 UT (not shown). We selected the subset of data including this observation for a detailed future study employing the PMC images and derived quantities. The successive mesospheric bores studied by Fritts et al. (2020) also belong to this group. Their signatures include extensions of the PMC layer to lower altitudes (in fact the lowest altitudes in the BOLIDE dataset), as demonstrated by the observations on 13 July 2018 in
Fig. 7bc. For a duration of few minutes only, the layer spreads to approximately its double width while the brightness remains





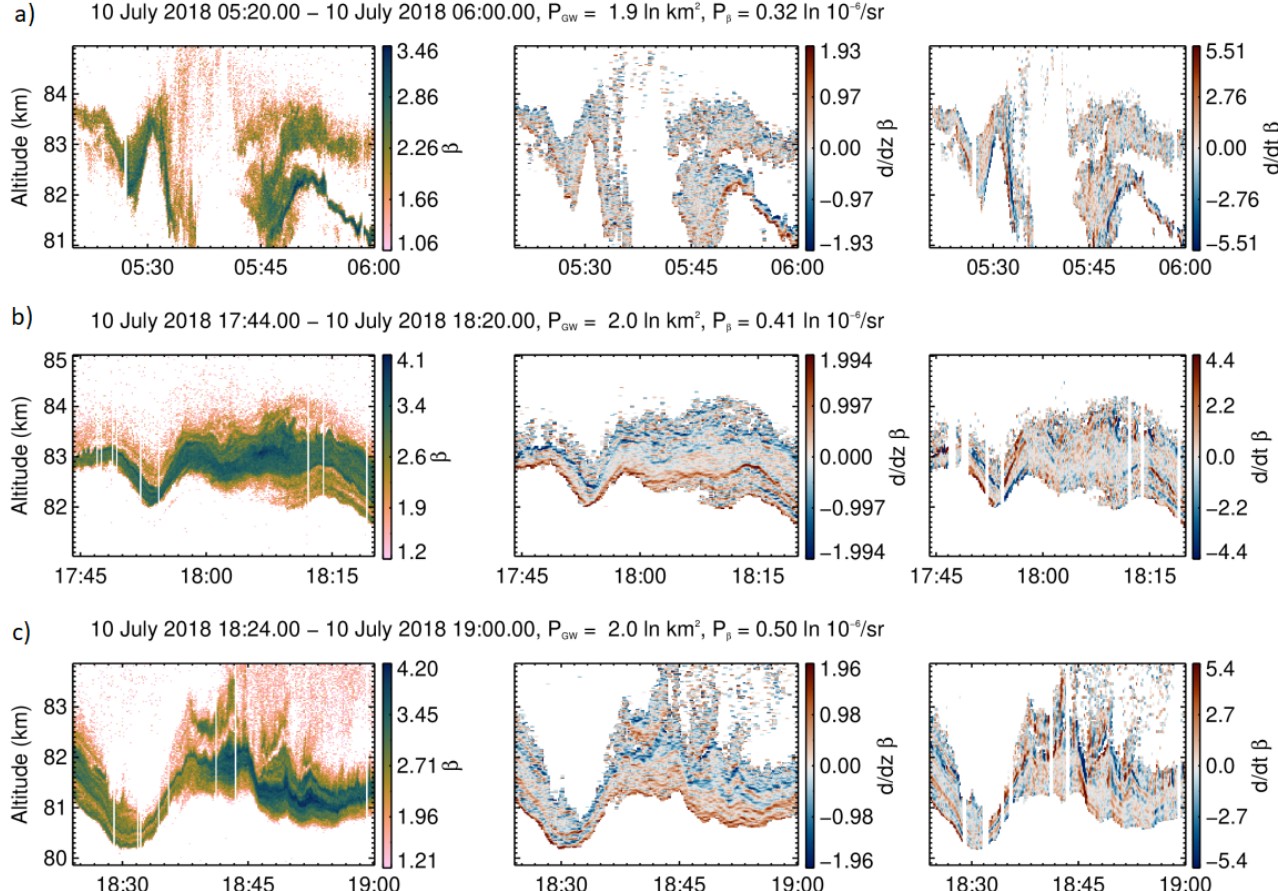

**Figure 6.** Selected PMC backscatter profiles which are indicative of gravity wave breaking in response to steepening of gravity waves. Same format as in Fig. 5.

about constant, leading to a strong increase in integrated brightness. These bores were accompanied by various and intense leading and trailing instabilities (Fritts et al., 2020). At the times of largest descent, tendrils associated with Kelvin-Helmholtz instabilities extrude from the PMC layer (Fig. 7b). Between 13:20 UT and 13:35 UT, localized increases in brightness indicate successions of small-scale vortex rings, as deduced from the joint analysis of images and lidar soundings. The observation and

identification of small-scale vortex rings in lidar data has first been accomplished using the BOLIDE lidar dataset.

4. PMC layers that show breakup and/or mixing of previously multi-layered, thin features. This class of events is sometimes accompanied by apparent breaking or overturning events associated with group 2. Examples are shown in Fig. 8. These events are ideal targets for modelling efforts of the effect of gravity waves on PMC layers, as thin layers can be regarded as reliable tracers of small-scale variability. As with group 2, this type of layers conform to the irregular or non-parallel multipe layer-

categories defined by Schäfer et al. (2020, their category IIIb and IIIcii), and make up for about 20 % of the BOLIDE dataset







**Figure 7.** PMC layers showing bore-like responses during periods with weak gravity wave activity. Same format as Fig. 5.

(13 h). All other BOLIDE data with profiles showing large gradients can be sorted into categories of short-period height variations in both narrow and wide layers defined by Schäfer et al. (2020, , their category Ib and IIbii).

Our analysis in Fig. 4 showed that large gradients are likely to occur during conditions of both strong and weak gravity wave activity. Most of the examples featured here belong to category B or D (large $P_{GW}$), but also PMC layers associated with low values of $P_{GW}$ can show distinct small-scale variability. The case with the lowest value is 11 July 10:30–12:30 UT,





**Figure 8.** Multi-layered PMC structures experiencing breakup or mixing. Same format as in Fig. 5.

a moderately-bright layer of 1-2 km thickness that contains a variety of internal structures. Although undulations in altitude are small during this period, there are large undulations before and after that were likely induced by a gravity wave with a period larger than 60 min. In conclusion, we observed small-scale variability in PMC layer in all conditions of weak, moderate and strong short-period gravity wave activity and the occurrence of truly unperturbed layers can be regarded as a rare



phenomenon. We find that PMC observations with below-threshold gradients are generally associated with wide, dim layers
      without discernible inner structure (category IIIa of Schäfer et al., 2020) or well defined thin layers of low brightness with
      distinct sinusoidal motions (our category B, category I of Schäfer et al., 2020). An example is a set of very narrow PMC layers
      with sinusoidal undulations of about 20 min period shown in Fig. 5b. While in the beginning there are several well-defined
      stacked layers, over time the brightness decreases and gradients become smaller. This behavious is in agreement with theoret-

ical results showing that, under conditions of gravity wave activity, weak layers are further mixed and ice particles are likely
      to sublimate, resulting in volume backscatter coefficients that are below the detection threshold. A comprehensive and general
      assessment of the effect of short-period gravity waves on the ice particle size and number density is likely difficult to obtain
      from lidar profiling data alone. Atmospheric dynamics occurs in three dimensions, and knowledge of the phase speeds and
      orientations of gravity waves relative to the wind speed and direction and thus the transport of ice particles is required for a

comprehensive analysis. In particular, the temporal axis we show in this work transforms via the local wind speed and the lidar
      beam movement across the sky to a spatial axis. Our resolution of 10 s likely relates to few hundred meters in the horizon-
      tal domain. The required information can be obtained by future experiments including e.g. coincident and common volume
      imaging and radar observations.

## 5   Conclusions

We set out to systematically screen, identify and characterize signatures of dynamical instabilites generated by the breaking of
      gravity waves and other nonlinear processes in PMC Turbo BOLIDE lidar data. The focus of this work was on the smallest
      scales (at 20 m and 10 s resolution), and we applied an easy-to-implement, general method to our lidar data without presuppos-
      ing any specific patterns. We presented statistics of gradients occurring in PMC backscatter data and performed a classification
      of PMC layers based on gradient thresholds, layer brightness and gravity wave activity. In accordance with visual observations,

we confirmed that smooth, unperturbed PMC layers are rare, and that small-scale structures are very common. We found that
      probability density functions of PMC brightness gradients exhibit longer tails compared to a normal distribution, an indication
      for intermittency. Large gradients in the vertical and temporal/horizontal dimension occur frequently during times with both
      weak and strong gravity wave activity. We believe the cases identified and presented in this work as well as the quantification
      of gradients will be helpful for future modelling of the effects of gravity waves on PMC layers.

*Data availability.*  The BOLIDE data are available from zenodo (Kaifler, 2021). A copy is hosted by NASA's Space Physics Data Facitily at
      https://cdaweb.gsfc.nasa.gov/ under section "Balloons".

*Author contributions.*  NK analyzed the data, prepared the figures and wrote the manuscript. BK designed and build the instrument, operated
it during flight and provided binned photon count data. MR obtained funding for the instrument. DF is the mission PI of PMC Turbo. All
authors contributed to the interpretation of the results and reviewed the manuscript.





*Competing interests.*  The authors declare no competing interests.

*Acknowledgements.*  We thank the PMC Turbo science team for discussion of the data analysis and results and review of the manuscript.





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
