# Peer review of "Signatures of gravity wave-induced instabilities in balloon lidar soundings of polar mesospheric clouds"

_Atmospheric Chemistry and Physics, 2022_

## Author Response (AR1)

We thank Referee #2 for the positive review and we appreciate the suggestions to improve the manuscript. Please find the point-by-point response below.

*l. 14/15: correct spelling "a unique"*

Done, thank you

*l. 34: missing word: should be "strong gravity wave activity" in the beginning of the line*

Done, thank you

*l. 44: I suggest formulating the connection between the sentence starting with "At timescales of minutes," and the previous sentences better. The connection content wise is clear, but in the current wording I am not sure if "the PMC layer" in l. 44 refers to the specific case from the previously mentioned study or is meant as a more general statement.*

This was a general statement, and we added "generally" in the sentence.

*l. 75: The lidar beam was tilted 28 degrees off-zenith, however, the abstract's first sentence tells about "near-vertical" profiles. Is 28 degrees off-zenith still considered near-vertical or does the statement in the abstract refer to profiles already converted from slant range to vertical range (if the latter, why "near" vertical)?*

Yes, 28 deg was considered "near-vertical". We tried to make the point that we analyze vertical profiles, that were however deduced from 28 deg-slanted measurements, using too few words. We have changed the expression to "vertical profiles" in the abstract in agreement with the rest of the text, as it is clear from section 2.1 that the beam was 28 deg off-zenith.

*l. 113/4: How long are the times of below-threshold or no PMC detections? Are they short enough so that linear interpolation is preferred to skipping these times?*

In total, the times of below-threshold or no PMC detections are about 68%. As seen in Kaifler, ESSD, 2021, their Fig. 6,7, PMC occur during a number of events that last several hours. During these (relevant) times, only very short gaps occur. To handle them, it is reasonable to interpolate in order to obtain useful P_GW, as demonstrated in Fig. 2. Please note that the interpolated sections are masked and not used for subsequent analysis, they are only used during the spectral analysis. For the longer gaps between the major events, this essentially conforms to a zero padding, which ensures that also at the beginning and end of the PMC events we get useful results.

*l. 110-131: I find the use of the term "high frequency" slightly unclear from reading this paragraph. I get that the focus spectral range is 5-62 min (corresponding to frequencies lower than the buoyancy period since the buoyancy period is below 5 min in the mesopause region). From the statement about Fig. 2b (l. 129) I also understand that exactly this range is called "high frequency" (both lower and higher frequencies are omitted in the blue curve). What is then meant by "retain high-frequency gravity waves above the buoyancy frequency only" (l. 116/7)? Please clarify. Confusion might just arise from shifting between the use of frequency/period and the meaning of above/below in terms of numbers.*

As the term "high-frequency" was only used three times and only in this subsection, we replaced the relevant sentences, and extended the explanation to make the motivation more clear. Changed to:

l. 111: "generated by the breaking of a short-period gravity wave"

l. 114: "We focus on short-period gravity waves above the buoyancy frequency of about 5min by selecting the spectral range of 5-62 min period."

l. 129: "and the filtered time series effectively captures the part of the motion in the desired range, excluding both the few-hours and the minute-scale perturbations, such that a good representation for the gravity wave activity we seek to characterize"

*l. 133: correct spelling "independent"*

done

*l. 139: The potential of emerging/fading in just 35s challenges the role of PMC particles as passive tracers. Can you quantify how rare such extremely large gradients are and such keep up the assumption of PMC particles being passive tracers at minute scales cited earlier (Fritts et al., 1993; Dong et al., 2021)?*

The occurrence rate of such large gradients can be assessed from the distributions shown in Fig. 3, and our conclusions are that they are not seldom. Yet we do believe PMC particles are passive tracers especially at the small scales, because we argue that those patterns arise mainly due to dynamical processes, and not so much microphysical processes. It must be remembered that the ice particles move through the (also moving) lidar beam, that is we do not observe ice particles at rest. The particles we observe over an amount of time are therefore different particles, and their arrangement in space that accounts for the patterns we observe is mainly due to changes in ice particle number density that is modulated by dynamics. Yet brightness changes over more than few minutes can also arise due to sublimation of ice particles. Geach et al., 2020, also discuss this aspect in their section 3.

*Fig. 2bc: Consider adding a legend in addition to the information given in the caption.*

Done, and Fig. 2c will be split in two subfigures for clarity. We had used one subfigure to save space, but to avoid confusion, it is better to use two subfigures, as the dimensions are different.

*Fig. 3a: There is an oscillation at the very top of the distribution around zero (solid line). Do you have an idea where this comes from and whether it can be considered an artefact or real?*

This was a very good observation. We looked into the matter and it is not real. Part of it can be resolved by changing the choice of histogram binning, such that 0 is not in between two but in the middle of one bin, i.e. by using an uneven number of bins in a symmetric interval instead of an even number. But still, when using a small binsize, the distribution is not perfectly smooth at the very top. This is because the volume backscatter coefficient is not perfectly continuous. This goes back to counting photons, i.e. because our raw data is discrete.

*Fig. 4: I suggest renaming the title of the x-axis to "l (number of standard deviations)", so that there is a connection between the use of "l" in the caption and the figure itself.*

This was done as suggested.

*l. 166: In my opinion the reader could benefit from a brief introduction of the structure of the Discussion chapter here. The chapter refers first to a number of already picked and published case studies from the dataset before putting the remaining cases into four groups. This division into already looked at and remaining events feels somehow arbitrary (though understandable for practical reasons) and needs introduction.*

We added:
"In the next section we will look into the morphology of the layers that exhibit large gradients
. We
will first discuss already published cases in the light of our results, and then proceed to descri
be four
general groups of PMC layer sections based on their morphology and discuss their link to the
dynamical processes that led to their formation."

*Table 3: Could you in the text (around l. 180) elaborate on how the mean category is calculated based on $P_{GW}$ and $P_{\beta}$?*

P_GW and P_beta were averaged for the time periods listed in Table 3, and the corresponding category determined as defined in Table 2. This is only an approximate result, as the time periods are hours long and the conditions may vary. We changed l. 180 to
"including an approximate category based on $P_\mathrm{GW}$ and $P_{\beta}$ averaged over the time period listed"

*Also, I guess the times on 10 July should be UT as well?*

Yes, we added "UT" to the two lines for 10 July, thank you

*l. 175: Please clarify what "such events" refers to.*

Exchanged by "gradients larger than 2 sigma".

*l. 219-221: After reading about the helpfulness of associate data described here (wind, imagery) I am left with the question to what degree this kind of data is available or not (wind no, image yes?). Probably a further interpretation is beyond the scope of this study, but the reader would appreciate a short note.*

The high-resolution PMC Turbo images are to be published on NASA's Space Physics Data Facility website and in the meantime are available from Bjorn Kjellstrand on request. Local wind speeds can be derived from them by tracking small-scale features. We added to the text: "This information can be deduced from the PMC Turbo images that will be available from N ASA's Space Physics Data Facility or on request."

*l. 222: correct spelling: ... activity ... "reduces" PMC brightness ...*

done

*l. 230/1: Is there any possibility to state already how the bright band in imagery relates to the transition that defines the described third category? Can you from the coincidence conclude that this sudden increase in thickness of the band visible in the lidar must have occurred on a*

*scale of several hundred kilometres as this is the scale of the band on the image? This might of course be part of the announced future study, but I am missing some kind of interpretation here once the image is already mentioned.*

The bright band in the image is identified as a mesospheric bore as it is similar to bores known from airglow images and e.g. Fritts et al. (2020). Characteristics are few km width, several hundred km spanwise extent, high brightness and trailing instabilities. The examples in Fig. 7 are to our knowledge the only vertical soundings of such phenomena. It is reasonable to assume that the bore front induces a widening of the layer, depending on the relation of the bore altitude to the PMC layer altitude to lower or higher altitudes or both. I would assume some variability across the spanwise extent of the bore, but the increase in PMC brightness seen in the image is likely to be due to an increase of layer width. We have extended the sentences in the text related to this example to:
"At 5:13~UT on 11 July 2018, the core of a PMC layer is both deflected upward and downwa rd by 2~km and 1~km, respectively, induced by the passage of a mesospheric bore front. The corresponding imagery reveals such a bore by showing a narrow and bright band that extends several hundred km across the camera field of view and moves through the lidar beam at 5:13 ~UT (not shown). The increase in PMC brightness at the bore front revealed by the images is therefore caused by up- and downdrafts that increase the PMC layer width, hinting that in this case the bore altitude coincided with the PMC layer altitude."

*l. 245: Please clarify if the number of 20% refers to group 4 only or group 2 and 4 together. Is this result comparable to the occurrences in Schäfer et al., 2020?*

Yes, it referred to both groups and the result is comparable. We changed the text to
"Layers of group 4 and group 2 make up for about 20\,\% of the BOLIDE dataset (13~h), and conform to the irregular or non-parallel multiple layer-
categories defined by \citet[][their category IIIb and IIIcii]{Schaefer2020}, that accounted for about 25\,\% of the ALOMAR dataset."

*l. 259: correct spelling "behaviour"*

done

*For several references the doi-links as url are not given correctly and one can therefore not click on them from the bibliography directly.*

We have checked again and hope that now all dois can be clicked on in the bibliograhy.

*l. 320: For the reference Geach et al., 2020, I suggest giving the doi of the finally published article (https://doi.org/10.1029/2020JD033038) and not the given doi linking to the manuscript in an open archive.*

We have updated all citations of preprints with their final citation.

We thank Referee #1 for his suggestions and questions and provide our answers below.

*Line 36, "By means of..." I think the author is trying to say that GW or other small-scale dynamics changes the altitudes of ice particles and their surrounding temperatures. Please consider to revise.*

We changed the sentence to
"The altitude and temperature variations induced by gravity wave or other small-scale dynamics can result in the rapid sublimation of ice particles."

*Line 60, How can the lidar measure the PMC brightness? The lidar measures the strength of the lidar echoes, so it is easy to understand the lidar can measure the "density" and thickness of the layer. But does the brightness also depend upon the size of the ice particle, which the lidar cannot measure? Please clarify?*

The volume backscatter coefficient, often termed "PMC brightness", that the lidar measures depends on both the ice particle density and the ice particle's size (Mie scattering). In l. 79 we wrote "The relevant physical quantity measured by the lidar is the volume backscatter coefficient beta, a measure for PMC brightness influenced by both the size and number density of ice particles within the observed volume." Few large particles will result in the same volume backscatter coefficient as many smaller particles, and it is impossible to distinguish using a single laser wavelength. This is the same as with naked-eye observations, which is probably why the term "brightness" is also applied to lidar measurements. More details on the volume backscatter coefficient is found in Kaifler et al., ESSD, 2022. On the scales considered in this study, the changes in volume backscatter coefficient, however, are mainly due to changes in number density, because changes in size occur on longer timescales. We have added:
"Lidar measurements of the changes of volume backscatter coefficient, also termed "PMC brightness", are at these scales therefore predominantly due to changes in density, and not in particle's size." And "volume backscatter coefficient or PMC brightness" in line 60.

*Line 88, it would be helpful to describe how this b is defined based on the lidar measurement. I know it is articulated in the other papers, but I think it can make the paper more reader friendly.*

This was partly described in 75-79 and we have completed this explanation for the correction of background and range and normalization to a standard density profile. This removes the Rayleigh part and results in the volume backscatter coefficient.

*Line 104, "Notable is the bright ...", this sentence reads a bit strange. Also, I think the "brightness" above is referring to the lidar's echo strength.*

We changed the sentence to "The lower boundary, that descends by 1 km from 82.2 km to 81.2 km, is of both very large $\beta$ and very large $d/dz \beta$."

*Line 127-128, this sentence seems to be incomplete. Please consider to revise.*

Changed to "We illustrate the derivation of PGW and Pbeta in Fig. 2 using a PMC measurement on 11 July 2018 including the part around 3:30 UT we ..

*Line 134, 135, the nv and np parameters are not defined. What are they?*

They were just to abbreviate the number of PMC detections (brightness values) and the number of PMC profiles, to later estimate the percentage of how many of them are associated with gradients in the tails of the distribution. As they were only used in three instances (l. 134, Table 2, and l. 180), we have replaced them with text.

*Figure 2. The differences between the reconstructed results and the observations are quite noticeable, may need some explanations.*

We had on purpose selected a range of scales (excluding the very short and very large scales) to construct the proxy representing specific gravity waves. Both curves were printed into Fig. 2b because they show the same quantity, not because we expect a very good agreement (which we don't). The text was modified to: "The mean PMC altitude $z\_c$ (Fig. 2b, black curve) is a good proxy for layer altitude and the filtered time series effectively captures the part of the motion in the desired range, excluding both the few-hours- and the minute-scale perturbations, such that it is a good representation for the gravity wave activity we seek to characterize (Fig. 2b, blue curve)."

*In addition, the distinction in Figure 2c may need more discussion.*

We are not sure what is meant by "the distinction". The two quantities were merely plotted into one subplot to save space. As the dimensions are different, we agree it is better to make two subplots c) and d), as both quantities have different meanings.

*Line 181-182, it may be a good idea to address how these results of category reflect Fritts' assessment of GW activities and forcing.*

We added: "The assessment by Fritts et al., 2019 of weak, moderate or strong dynamics or forcing is reflected in our result for category, in the way that mostly weak dynamics or forcing relates to category A or C, and stronger dynamics and forcing to category B."

*Line 202-204, what is "high definition of the layer edges"? In addition, "Second, due to…" this sentence reads funny. Please consider to revise.*

Rephrased
to: "This type of PMC layers have sharp boundaries, often at the lower edge, and thus vertical gradients are large where the layers emerge from or fade into the background. This is seen e.g. in $\partial\_z\beta$ on 10 July 2018 at 19:20 -- 20:10~UT (Fig. 5a). Because of the oscillatory motions of these layers, also large temporal gradients are induced, e.g. on 13 July 2018 10:25--11:05~UT (Fig. 5c).

*Line 207, it may be helpful to show some PMC images that are associated with the discussion here to provide the direct evidence of your argument.*

We have added the reference to Fritts et al., 2019, where high-resolution images from these times are shown and interpreted. A more detailed study of single cases will be done during future work.

*Line 218, I am not sure the feature can indicate "instability dynamics", without the in situ temperature and horizontal wind information, just physical form of the lidar echoes may implicate some instabilities, but not enough to draw this conclusion. The same can be said for*

*the conclusion, the first sentence in the Conclusion. These results may be the indirect evidences of the atmospheric instability.*

l. 218 was changed to "likely caused by.." We had mentioned in l. 219 that additional information is necessary. The first line in the conclusions was changed to "small-scale signatures that are likely caused by.."